# Stochastic Majorization-Minimization Algorithms for Large-Scale Optimization

**Julien Mairal**
LEAR Project-Team - INRIA Grenoble
`julien.mairal@inria.fr`

## Abstract

Majorization-minimization algorithms consist of iteratively minimizing a majorizing surrogate of an objective function. Because of its simplicity and its wide applicability, this principle has been very popular in statistics and in signal processing. In this paper, we intend to make this principle scalable. We introduce a stochastic majorization-minimization scheme which is able to deal with large-scale or possibly infinite data sets. When applied to convex optimization problems under suitable assumptions, we show that it achieves an expected convergence rate of $O(1/\sqrt{n})$ after $n$ iterations, and of $O(1/n)$ for strongly convex functions. Equally important, our scheme almost surely converges to stationary points for a large class of non-convex problems. We develop several efficient algorithms based on our framework. First, we propose a new stochastic proximal gradient method, which experimentally matches state-of-the-art solvers for large-scale $\ell_1$-logistic regression. Second, we develop an online DC programming algorithm for non-convex sparse estimation. Finally, we demonstrate the effectiveness of our approach for solving large-scale structured matrix factorization problems.

## 1 Introduction

Majorization-minimization [15] is a simple optimization principle for minimizing an objective function. It consists of iteratively minimizing a surrogate that upper-bounds the objective, thus monotonically driving the objective function value downhill. This idea is used in many existing procedures. For instance, the expectation-maximization (EM) algorithm (see [5, 21]) builds a surrogate for a likelihood model by using Jensen's inequality. Other approaches can also be interpreted under the majorization-minimization point of view, such as DC programming [8], where "DC" stands for difference of convex functions, variational Bayes techniques [28], or proximal algorithms [1, 23, 29].

In this paper, we propose a stochastic majorization-minimization algorithm, which is is suitable for solving large-scale problems arising in machine learning and signal processing. More precisely, we address the minimization of an expected cost—that is, an objective function that can be represented by an expectation over a data distribution. For such objectives, online techniques based on stochastic approximations have proven to be particularly efficient, and have drawn a lot of attraction in machine learning, statistics, and optimization [3–6, 9–12, 14, 16, 17, 19, 22, 24–26, 30].

Our scheme follows this line of research. It consists of iteratively building a surrogate of the expected cost when only a single data point is observed at each iteration; this data point is used to update the surrogate, which in turn is minimized to obtain a new estimate. Some previous works are closely related to this scheme: the online EM algorithm for latent data models [5, 21] and the online matrix factorization technique of [19] involve for instance surrogate functions updated in a similar fashion. Compared to these two approaches, our method is targeted to more general optimization problems.

Another related work is the incremental majorization-minimization algorithm of [18] for finite training sets; it was indeed shown to be efficient for solving machine learning problems where storing

dense information about the past iterates can be afforded. Concretely, this incremental scheme requires to store $O(pn)$ values, where $p$ is the variable size, and $n$ is the size of the training set.[1] This issue was the main motivation for us for proposing a stochastic scheme with a memory load independent of $n$, thus allowing us to possibly deal with infinite data sets, or a huge variable size $p$.

We study the convergence properties of our algorithm when the surrogates are strongly convex and chosen among the class of *first-order surrogate functions* introduced in [18], which consist of approximating the possibly non-smooth objective up to a smooth error. When the objective is convex, we obtain expected convergence rates that are asymptotically optimal, or close to optimal [14, 22]. More precisely, the convergence rate is of order $O(1/\sqrt{n})$ in a finite horizon setting, and $O(1/n)$ for a strongly convex objective in an infinite horizon setting. Our second analysis shows that for *non-convex* problems, our method almost surely converges to a set of stationary points under suitable assumptions. We believe that this result is equally valuable as convergence rates for convex optimization. To the best of our knowledge, the literature on stochastic non-convex optimization is rather scarce, and we are only aware of convergence results in more restricted settings than ours—see for instance [3] for the stochastic gradient descent algorithm, [5] for online EM, [19] for online matrix factorization, or [9], which provides stronger guarantees, but for unconstrained smooth problems.

We develop several efficient algorithms based on our framework. The first one is a new stochastic proximal gradient method for composite or constrained optimization. This algorithm is related to a long series of work in the convex optimization literature [6, 10, 12, 14, 16, 22, 25, 30], and we demonstrate that it performs as well as state-of-the-art solvers for large-scale $\ell_1$-logistic regression [7]. The second one is an online DC programming technique, which we demonstrate to be better than batch alternatives for large-scale non-convex sparse estimation [8]. Finally, we show that our scheme can address efficiently structured sparse matrix factorization problems in an online fashion, and offers new possibilities to [13, 19] such as the use of various loss or regularization functions.

This paper is organized as follows: Section 2 introduces first-order surrogate functions for batch optimization; Section 3 is devoted to our stochastic approach and its convergence analysis; Section 4 presents several applications and numerical experiments, and Section 5 concludes the paper.

## 2 Optimization with First-Order Surrogate Functions

Throughout the paper, we are interested in the minimization of a continuous function $f : \mathbb{R}^p \to \mathbb{R}$:

$$\min_{\theta \in \Theta} f(\theta), \tag{1}$$

where $\Theta \subseteq \mathbb{R}^p$ is a convex set. The majorization-minimization principle consists of computing a majorizing surrogate $g_n$ of $f$ at iteration $n$ and updating the current estimate by $\theta_n \in \arg\min_{\theta \in \Theta} g_n(\theta)$. The success of such a scheme depends on how well the surrogates approximate $f$. In this paper, we consider a particular class of surrogate functions introduced in [18] and defined as follows:

**Definition 2.1 (Strongly Convex First-Order Surrogate Functions).**
*Let $\kappa$ be in $\Theta$. We denote by $\mathcal{S}_{L,\rho}(f, \kappa)$ the set of $\rho$-strongly convex functions $g$ such that $g \geq f$, $g(\kappa) = f(\kappa)$, the approximation error $g - f$ is differentiable, and the gradient $\nabla(g - f)$ is $L$-Lipschitz continuous. We call the functions $g$ in $\mathcal{S}_{L,\rho}(f, \kappa)$ "first-order surrogate functions".*

Among the first-order surrogate functions presented in [18], we should mention the following ones:
• **Lipschitz Gradient Surrogates.**
When $f$ is differentiable and $\nabla f$ is $L$-Lipschitz, $f$ admits the following surrogate $g$ in $\mathcal{S}_{2L,L}(f, \kappa)$:

$$g : \theta \mapsto f(\kappa) + \nabla f(\kappa)^\top (\theta - \kappa) + \frac{L}{2}\|\theta - \kappa\|_2^2.$$

When $f$ is convex, $g$ is in $\mathcal{S}_{L,L}(f, \kappa)$, and when $f$ is $\mu$-strongly convex, $g$ is in $\mathcal{S}_{L-\mu,L}(f, \kappa)$. Minimizing $g$ amounts to performing a classical classical gradient descent step $\theta \leftarrow \kappa - \frac{1}{L}\nabla f(\kappa)$.
• **Proximal Gradient Surrogates.**
Assume that $f$ splits into $f = f_1 + f_2$, where $f_1$ is differentiable, $\nabla f_1$ is $L$-Lipschitz, and $f_2$ is

convex. Then, the function $g$ below is in $\mathcal{S}_{2L,L}(f,\kappa)$:

$$g : \theta \mapsto f_1(\kappa) + \nabla f_1(\kappa)^\top(\theta - \kappa) + \frac{L}{2}\|\theta - \kappa\|_2^2 + f_2(\theta).$$

When $f_1$ is convex, $g$ is in $\mathcal{S}_{L,L}(f,\kappa)$. If $f_1$ is $\mu$-strongly convex, $g$ is in $\mathcal{S}_{L-\mu,L}(f,\kappa)$. Minimizing $g$ amounts to a proximal gradient step [1, 23, 29]: $\theta \leftarrow \arg\min_\theta \frac{1}{2}\|\kappa - \frac{1}{L}\nabla f_1(\kappa) - \theta\|_2^2 + \frac{1}{L}f_2(\theta)$.

• **DC Programming Surrogates.**
Assume that $f = f_1 + f_2$, where $f_2$ is concave and differentiable, $\nabla f_2$ is $L_2$-Lipschitz, and $g_1$ is in $\mathcal{S}_{L_1,\rho_1}(f_1,\kappa)$, Then, the following function $g$ is a surrogate in $\mathcal{S}_{L_1+L_2,\rho_1}(f,\kappa)$:

$$g : \theta \mapsto f_1(\theta) + f_2(\kappa) + \nabla f_2(\kappa)^\top(\theta - \kappa).$$

When $f_1$ is convex, $f_1 + f_2$ is a difference of convex functions, leading to a DC program [8].

With the definition of first-order surrogates and a basic "batch" algorithm in hand, we now introduce our main contribution: a stochastic scheme for solving large-scale problems.

## 3 Stochastic Optimization

As pointed out in [4], one is usually not interested in the minimization of an *empirical cost* on a finite training set, but instead in minimizing an *expected cost*. Thus, we assume from now on that $f$ has the form of an expectation:

$$\min_{\theta \in \Theta} \left[ f(\theta) \triangleq \mathbb{E}_{\mathbf{x}}[\ell(\mathbf{x}, \theta)] \right], \tag{2}$$

where $\mathbf{x}$ from some set $\mathcal{X}$ represents a data point, which is drawn according to some unknown distribution, and $\ell$ is a continuous loss function. As often done in the literature [22], we assume that the expectations are well defined and finite valued; we also assume that $f$ is bounded below.

We present our approach for tackling (2) in Algorithm 1. At each iteration, we draw a training point $\mathbf{x}_n$, assuming that these points are i.i.d. samples from the data distribution. Note that in practice, since it is often difficult to obtain true i.i.d. samples, the points $\mathbf{x}_n$ are computed by cycling on a randomly permuted training set [4]. Then, we choose a surrogate $g_n$ for the function $\theta \mapsto \ell(\mathbf{x}_n, \theta)$, and we use it to update a function $\bar{g}_n$ that behaves as an approximate surrogate for the expected cost $f$. The function $\bar{g}_n$ is in fact a weighted average of previously computed surrogates, and involves a sequence of weights $(w_n)_{n \geq 1}$ that will be discussed later. Then, we minimize $\bar{g}_n$, and obtain a new estimate $\theta_n$. For convex problems, we also propose to use averaging schemes, denoted by "option 2" and "option 3" in Alg. 1. Averaging is a classical technique for improving convergence rates in convex optimization [10, 22] for reasons that are clear in the convergence proofs.

---

**Algorithm 1** Stochastic Majorization-Minimization Scheme

**input** $\theta_0 \in \Theta$ (initial estimate); $N$ (number of iterations); $(w_n)_{n \geq 1}$, weights in $(0, 1]$;
1: initialize the approximate surrogate: $\bar{g}_0 : \theta \mapsto \frac{\rho}{2}\|\theta - \theta_0\|_2^2$; $\bar{\theta}_0 = \theta_0$; $\hat{\theta}_0 = \theta_0$;
2: **for** $n = 1, \ldots, N$ **do**
3:     draw a training point $\mathbf{x}_n$; define $f_n : \theta \mapsto \ell(\mathbf{x}_n, \theta)$;
4:     choose a surrogate function $g_n$ in $\mathcal{S}_{L,\rho}(f_n, \theta_{n-1})$;
5:     update the approximate surrogate: $\bar{g}_n = (1 - w_n)\bar{g}_{n-1} + w_n g_n$;
6:     update the current estimate:

$$\theta_n \in \arg\min_{\theta \in \Theta} \bar{g}_n(\theta);$$

7:     for option 2, update the averaged iterate: $\hat{\theta}_n \triangleq (1 - w_{n+1})\hat{\theta}_{n-1} + w_{n+1}\theta_n$;
8:     for option 3, update the averaged iterate: $\bar{\theta}_n \triangleq \frac{(1-w_{n+1})\bar{\theta}_{n-1} + w_{n+1}\theta_n}{\sum_{k=1}^{n+1} w_k}$;
9: **end for**
**output (option 1):** $\theta_N$ (current estimate, no averaging);
**output (option 2):** $\bar{\theta}_N$ (first averaging scheme);
**output (option 3):** $\hat{\theta}_N$ (second averaging scheme).

---

We remark that Algorithm 1 is only practical when the functions $\bar{g}_n$ can be parameterized with a small number of variables, and when they can be easily minimized over $\Theta$. Concrete examples are discussed in Section 4. Before that, we proceed with the convergence analysis.

## 3.1 Convergence Analysis - Convex Case

First, We study the case of convex functions $f_n : \theta \mapsto \ell(\theta, \mathbf{x}_n)$, and make the following assumption:

**(A)** for all $\theta$ in $\Theta$, the functions $f_n$ are $R$-Lipschitz continuous. Note that for convex functions, this is equivalent to saying that subgradients of $f_n$ are uniformly bounded by $R$.

Assumption **(A)** is classical in the stochastic optimization literature [22]. Our first result shows that with the averaging scheme corresponding to "option 2" in Alg. 1, we obtain an expected convergence rate that makes explicit the role of the weight sequence $(w_n)_{n \geq 1}$.

**Proposition 3.1** (**Convergence Rate**).
*When the functions $f_n$ are convex, under assumption (A), and when $\rho = L$, we have*

$$\mathbb{E}[f(\bar{\theta}_{n-1}) - f^\star] \leq \frac{L\|\theta^\star - \theta_0\|_2^2 + \frac{R^2}{L}\sum_{k=1}^n w_k^2}{2\sum_{k=1}^n w_k} \qquad \text{for all } n \geq 1, \tag{3}$$

*where $\bar{\theta}_{n-1}$ is defined in Algorithm 1, $\theta^\star$ is a minimizer of $f$ on $\Theta$, and $f^\star \triangleq f(\theta^\star)$.*

Such a rate is similar to the one of stochastic gradient descent with averaging, see [22] for example. Note that the constraint $\rho = L$ here is compatible with the proximal gradient surrogate.

From Proposition 3.1, it is easy to obtain a $O(1/\sqrt{n})$ bound for a finite horizon—that is, when the total number of iterations $n$ is known in advance. When $n$ is fixed, such a bound can indeed be obtained by plugging constant weights $w_k = \gamma/\sqrt{n}$ for all $k \leq n$ in Eq. (3). Note that the upper-bound $O(1/\sqrt{n})$ cannot be improved in general without making further assumptions on the objective function [22]. The next corollary shows that in an infinite horizon setting and with decreasing weights, we lose a logarithmic factor compared to an optimal convergence rate [14,22] of $O(1/\sqrt{n})$.

**Corollary 3.1** (**Convergence Rate - Infinite Horizon - Decreasing Weights**).
*Let us make the same assumptions as in Proposition 3.1 and choose the weights $w_n = \gamma/\sqrt{n}$. Then,*

$$\mathbb{E}[f(\bar{\theta}_{n-1}) - f^\star] \leq \frac{L\|\theta^\star - \theta_0\|_2^2}{2\gamma\sqrt{n}} + \frac{R^2\gamma(1 + \log(n))}{2L\sqrt{n}}, \quad \forall n \geq 2.$$

Our analysis suggests to use weights of the form $O(1/\sqrt{n})$. In practice, we have found that choosing $w_n = \sqrt{n_0 + 1}/\sqrt{n_0 + n}$ performs well, where $n_0$ is tuned on a subsample of the training set.

## 3.2 Convergence Analysis - Strongly Convex Case

In this section, we introduce an additional assumption:

**(B)** the functions $f_n$ are $\mu$-strongly convex.

We show that our method achieves a rate $O(1/n)$, which is optimal up to a multiplicative constant for strongly convex functions (see [14, 22]).

**Proposition 3.2** (**Convergence Rate**).
*Under assumptions (A) and (B), with $\rho = L + \mu$. Define $\beta \triangleq \frac{\mu}{\rho}$ and $w_n \triangleq \frac{1+\beta}{1+\beta n}$. Then,*

$$\mathbb{E}[f(\hat{\theta}_{n-1}) - f^\star] + \frac{\rho}{2}\mathbb{E}[\|\theta^\star - \theta_n\|_2^2] \leq \max\left(\frac{2R^2}{\mu}, \rho\|\theta^\star - \theta_0\|_2^2\right)\frac{1}{\beta n + 1} \quad \text{for all } n \geq 1,$$

*where $\hat{\theta}_n$ is defined in Algorithm 1, when choosing the averaging scheme called "option 3".*

The averaging scheme is slightly different than in the previous section and the weights decrease at a different speed. Again, this rate applies to the proximal gradient surrogates, which satisfy the constraint $\rho = L + \mu$. In the next section, we analyze our scheme in a non-convex setting.

## 3.3 Convergence Analysis - Non-Convex Case

Convergence results for non-convex problems are by nature weak, and difficult to obtain for stochastic optimization [4, 9]. In such a context, proving convergence to a global (or local) minimum is out of reach, and classical analyses study instead asymptotic stationary point conditions, which involve directional derivatives (see [2, 18]). Concretely, we introduce the following assumptions:

**(C)** $\Theta$ and the support $\mathcal{X}$ of the data are compact;

**(D)** The functions $f_n$ are uniformly bounded by some constant $M$;

**(E)** The weights $w_n$ are non-increasing, $w_1 = 1$, $\sum_{n \geq 1} w_n = +\infty$, and $\sum_{n \geq 1} w_n^2 \sqrt{n} < +\infty$;

**(F)** The directional derivatives $\nabla f_n(\theta, \theta' - \theta)$, and $\nabla f(\theta, \theta' - \theta)$ exist for all $\theta$ and $\theta'$ in $\Theta$.

Assumptions **(C)** and **(D)** combined with **(A)** are useful because they allow us to use some uniform convergence results from the theory of empirical processes [27]. In a nutshell, these assumptions ensure that the function class $\{\mathbf{x} \mapsto \ell(\mathbf{x}, \theta) : \theta \in \Theta\}$ is "simple enough", such that a uniform law of large numbers applies. The assumption **(E)** is more technical: it resembles classical conditions used for proving the convergence of stochastic gradient descent algorithms, usually stating that the weights $w_n$ should be the summand of a diverging sum while the sum of $w_n^2$ should be finite; the constraint $\sum_{n \geq 1} w_n^2 \sqrt{n} < +\infty$ is slightly stronger. Finally, **(F)** is a mild assumption, which is useful to characterize the stationary points of the problem. A classical necessary first-order condition [2] for $\theta$ to be a local minimum of $f$ is indeed to have $\nabla f(\theta, \theta' - \theta)$ non-negative for all $\theta'$ in $\Theta$. We call such points $\theta$ the stationary points of the function $f$. The next proposition is a generalization of a convergence result obtained in [19] in the context of sparse matrix factorization.

**Proposition 3.3** (**Non-Convex Analysis - Almost Sure Convergence**).
*Under assumptions (A), (C), (D), (E), $(f(\theta_n))_{n \geq 0}$ converges with probability one. Under assumption (F), we also have that*

$$\liminf_{n \to +\infty} \inf_{\theta \in \Theta} \frac{\nabla \bar{f}_n(\theta_n, \theta - \theta_n)}{\|\theta - \theta_n\|_2} \geq 0,$$

*where the function $\bar{f}_n$ is a weighted empirical risk recursively defined as $\bar{f}_n = (1 - w_n)\bar{f}_{n-1} + w_n f_n$. It can be shown that $\bar{f}_n$ uniformly converges to $f$.*

Even though $\bar{f}_n$ converges uniformly to the expected cost $f$, Proposition 3.3 does not imply that the limit points of $(\theta_n)_{n \geq 1}$ are stationary points of $f$. We obtain such a guarantee when the surrogates that are parameterized, an assumption always satisfied when Algorithm 1 is used in practice.

**Proposition 3.4** (**Non-Convex Analysis - Parameterized Surrogates**).
*Let us make the same assumptions as in Proposition 3.3, and let us assume that the functions $\bar{g}_n$ are parameterized by some variables $\kappa_n$ living in a compact set $\mathcal{K}$ of $\mathbb{R}^d$. In other words, $\bar{g}_n$ can be written as $g_{\kappa_n}$, with $\kappa_n$ in $\mathcal{K}$. Suppose there exists a constant $K > 0$ such that $|g_\kappa(\theta) - g_{\kappa'}(\theta)| \leq K\|\kappa - \kappa'\|_2$ for all $\theta$ in $\Theta$ and $\kappa, \kappa'$ in $\mathcal{K}$. Then, every limit point $\theta_\infty$ of the sequence $(\theta_n)_{n \geq 1}$ is a stationary point of $f$—that is, for all $\theta$ in $\Theta$,*

$$\nabla f(\theta_\infty, \theta - \theta_\infty) \geq 0.$$

Finally, we show that our non-convex convergence analysis can be extended beyond first-order surrogate functions—that is, when $g_n$ does not satisfy exactly Definition 2.1. This is possible when the objective has a particular partially separable structure, as shown in the next proposition. This extension was motivated by the non-convex sparse estimation formulation of Section 4, where such a structure appears.

**Proposition 3.5** (**Non-Convex Analysis - Partially Separable Extension**).
*Assume that the functions $f_n$ split into $f_n(\theta) = f_{0,n}(\theta) + \sum_{k=1}^K f_{k,n}(\gamma_k(\theta))$, where the functions $\gamma_k : \mathbb{R}^p \to \mathbb{R}$ are convex and $R$-Lipschitz, and the $f_{k,n}$ are non-decreasing for $k \geq 1$. Consider $g_{n,0}$ in $\mathcal{S}_{L_0, \rho_1}(f_{0,n}, \theta_{n-1})$, and some non-decreasing functions $g_{k,n}$ in $\mathcal{S}_{L_k, 0}(f_{k,n}, \gamma_k(\theta_{n-1}))$. Instead of choosing $g_n$ in $\mathcal{S}_{L,\rho}(f_n, \theta_{n-1})$ in Alg 1, replace it by $g_n \triangleq \theta \mapsto g_{0,n}(\theta) + g_{k,n}(\gamma_k(\theta))$.*

*Then, Propositions 3.3 and 3.4 still hold.*

## 4    Applications and Experimental Validation

In this section, we introduce different applications, and provide numerical experiments. A C++/Matlab implementation is available in the software package SPAMS [19].[2] All experiments were performed on a single core of a 2GHz Intel CPU with $64$GB of RAM.

---

## 4.1 Stochastic Proximal Gradient Descent Algorithm

Our first application is a stochastic proximal gradient descent method, which we call SMM (Stochastic Majorization-Minimization), for solving problems of the form:

$$\min_{\theta \in \Theta} \mathbb{E}_{\mathbf{x}}[\ell(\mathbf{x}, \theta)] + \psi(\theta), \tag{4}$$

where $\psi$ is a convex deterministic regularization function, and the functions $\theta \mapsto \ell(\mathbf{x}, \theta)$ are differentiable and their gradients are $L$-Lipschitz continuous. We can thus use the proximal gradient surrogate presented in Section 2. Assume that a weight sequence $(w_n)_{n \geq 1}$ is chosen such that $w_1 = 1$. By defining some other weights $w_n^i$ recursively as $w_n^i \triangleq (1 - w_n)w_n^{i-1}$ for $i < n$ and $w_n^n \triangleq w_n$, our scheme yields the update rule:

$$\theta_n \leftarrow \underset{\theta \in \Theta}{\arg\min} \sum_{i=1}^{n} w_n^i \left[ \nabla f_i(\theta_{i-1})^\top \theta + \frac{L}{2} \|\theta - \theta_{i-1}\|_2^2 + \psi(\theta) \right]. \tag{SMM}$$

Our algorithm is related to FOBOS [6], to SMIDAS [25] or the truncated gradient method [16] (when $\psi$ is the $\ell_1$-norm). These three algorithms use indeed the following update rule:

$$\theta_n \leftarrow \underset{\theta \in \Theta}{\arg\min} \nabla f_n(\theta_{n-1})^\top \theta + \frac{1}{2\eta_n} \|\theta - \theta_{n-1}\|_2^2 + \psi(\theta), \tag{FOBOS}$$

Another related scheme is the regularized dual averaging (RDA) of [30], which can be written as

$$\theta_n \leftarrow \underset{\theta \in \Theta}{\arg\min} \frac{1}{n} \sum_{i=1}^{n} \nabla f_i(\theta_{i-1})^\top \theta + \frac{1}{2\eta_n} \|\theta\|_2^2 + \psi(\theta). \tag{RDA}$$

Compared to these approaches, our scheme includes a weighted average of previously seen gradients, and a weighted average of the past iterates. Some links can also be drawn with approaches such as the "approximate follow the leader" algorithm of [10] and other works [12, 14].

We now evaluate the performance of our method for $\ell_1$-logistic regression. In summary, the datasets consist of pairs $(y_i, \mathbf{x}_i)_{i=1}^N$, where the $y_i$'s are in $\{-1, +1\}$, and the $\mathbf{x}_i$'s are in $\mathbb{R}^p$ with unit $\ell_2$-norm. The function $\psi$ in (4) is the $\ell_1$-norm: $\psi(\theta) \triangleq \lambda \|\theta\|_1$, and $\lambda$ is a regularization parameter; the functions $f_i$ are logistic losses: $f_i(\theta) \triangleq \log(1 + e^{-y_i \mathbf{x}_i^\top \theta})$. One part of each dataset is devoted to training, and another part to testing. We used weights of the form $w_n \triangleq \sqrt{(n_0 + 1)/(n + n_0)}$, where $n_0$ is automatically adjusted at the beginning of each experiment by performing one pass on $5\%$ of the training data. We implemented SMM in C++ and exploited the sparseness of the datasets, such that each update has a computational complexity of the order $O(s)$, where $s$ is the number of non zeros in $\nabla f_n(\theta_{n-1})$; such an implementation is non trivial but proved to be very efficient.

We consider three datasets described in the table below. rcv1 and webspam are obtained from the 2008 Pascal large-scale learning challenge.[3] kdd2010 is available from the LIBSVM website.[4]

| name | $N_{\text{tr}}$ (train) | $N_{\text{te}}$ (test) | $p$ | density (%) | size (GB) |
|---|---|---|---|---|---|
| rcv1 | 781 265 | 23 149 | 47 152 | 0.161 | 0.95 |
| webspam | 250 000 | 100 000 | 16 091 143 | 0.023 | 14.95 |
| kdd2010 | 10 000 000 | 9 264 097 | 28 875 157 | $10^{-4}$ | 4.8 |

We compare our implementation with state-of-the-art publicly available solvers: the batch algorithm FISTA of [1] implemented in the C++ SPAMS toolbox and LIBLINEAR v1.93 [7]. LIBLINEAR is based on a working-set algorithm, and, to the best of our knowledge, is one of the most efficient available solver for $\ell_1$-logistic regression with sparse datasets. Because $p$ is large, the incremental majorization-minimization method of [18] could not run for memory reasons. We run every method on $1, 2, 3, 4, 5, 10$ and $25$ epochs (passes over the training set), for three regularization regimes, respectively yielding a solution with approximately $100, 1\,000$ and $10\,000$ non-zero coefficients. We report results for the medium regularization in Figure 1 and provide the rest as supplemental material. FISTA is not represented in this figure since it required more than $25$ epochs to achieve reasonable values. Our conclusion is that *SMM often provides a reasonable solution after one epoch, and outperforms LIBLINEAR in the low-precision regime. For high-precision regimes, LIBLINEAR should be preferred.* Such a conclusion is often obtained when comparing batch and stochastic algorithms [4], but matching the performance of LIBLINEAR is very challenging.

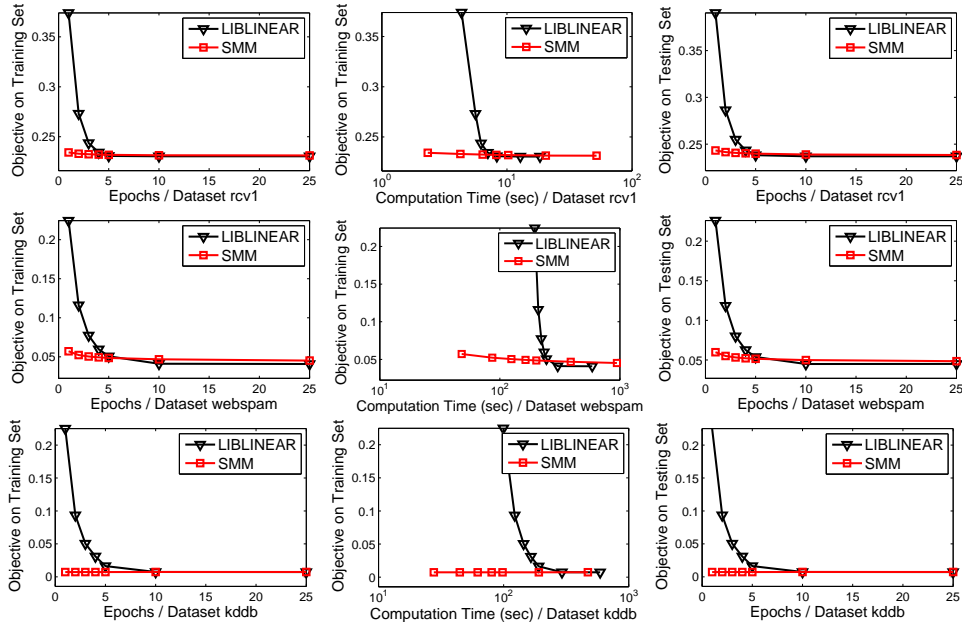

Figure 1: Comparison between LIBLINEAR and SMM for the medium regularization regime.

## 4.2 Online DC Programming for Non-Convex Sparse Estimation

We now consider the same experimental setting as in the previous section, but with a non-convex regularizer $\psi : \theta \mapsto \lambda \sum_{j=1}^p \log(|\theta[j]| + \varepsilon)$, where $\theta[j]$ is the $j$-th entry in $\theta$. A classical way for minimizing the regularized empirical cost $\frac{1}{N} \sum_{i=1}^N f_i(\theta) + \psi(\theta)$ is to resort to DC programming. It consists of solving a sequence of reweighted-$\ell_1$ problems [8]. A current estimate $\theta_{n-1}$ is updated as a solution of $\min_{\theta \in \Theta} \frac{1}{N} \sum_{i=1}^N f_i(\theta) + \lambda \sum_{j=1}^p \eta_j |\theta[j]|$, where $\eta_j \triangleq 1/(|\theta_{n-1}[j]| + \varepsilon)$.

In contrast to this "batch" methodology, we can use our framework to address the problem online. At iteration $n$ of Algorithm 1, we define the function $g_n$ according to Proposition 3.5:

$$g_n : \theta \mapsto f_n(\theta_{n-1}) + \nabla f_n(\theta_{n-1})^\top (\theta - \theta_{n-1}) + \frac{L}{2} \|\theta - \theta_{n-1}\|_2^2 + \lambda \sum_{j=1}^p \frac{|\theta[j]|}{|\theta_{n-1}[j]| + \varepsilon},$$

We compare our online DC programming algorithm against the batch one, and report the results in Figure 2, with $\varepsilon$ set to $0.01$. We conclude that *the batch reweighted-$\ell_1$ algorithm always converges after 2 or 3 weight updates, but suffers from local minima issues. The stochastic algorithm exhibits a slower convergence, but provides significantly better solutions.* Whether or not there are good theoretical reasons for this fact remains to be investigated. Note that it would have been more rigorous to choose a bounded set $\Theta$, which is required by Proposition 3.5. In practice, it turns not to be necessary for our method to work well; the iterates $\theta_n$ have indeed remained in a bounded set.

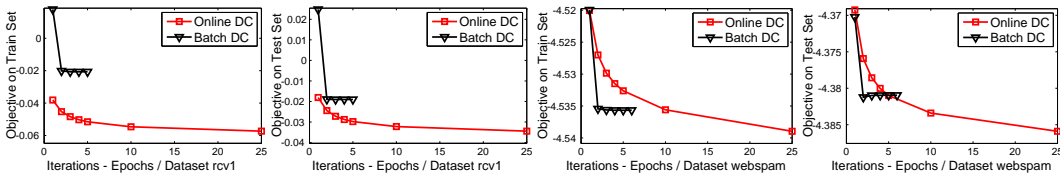

Figure 2: Comparison between batch and online DC programming, with medium regularization for the datasets rcv1 and webspam. Additional plots are provided in the supplemental material. Note that each iteration in the batch setting can perform several epochs (passes over training data).

## 4.3 Online Structured Sparse Coding

In this section, we show that we can bring new functionalities to existing matrix factorization techniques [13, 19]. We are given a large collection of signals $(\mathbf{x}_i)_{i=1}^N$ in $\mathbb{R}^m$, and we want to find a

dictionary $\mathbf{D}$ in $\mathbb{R}^{m \times K}$ that can represent these signals in a sparse way. The quality of $\mathbf{D}$ is measured through the loss $\ell(\mathbf{x}, \mathbf{D}) \triangleq \min_{\boldsymbol{\alpha} \in \mathbb{R}^K} \frac{1}{2}\|\mathbf{x} - \mathbf{D}\boldsymbol{\alpha}\|_2^2 + \lambda_1\|\boldsymbol{\alpha}\|_1 + \frac{\lambda_2}{2}\|\boldsymbol{\alpha}\|_2^2$, where the $\ell_1$-norm can be replaced by any convex regularizer, and the squared loss by any convex smooth loss.

Then, we are interested in minimizing the following expected cost:

$$\min_{\mathbf{D} \in \mathbb{R}^{m \times K}} \mathbb{E}_{\mathbf{x}}\left[\ell(\mathbf{x}, \mathbf{D})\right] + \varphi(\mathbf{D}),$$

where $\varphi$ is a regularizer for $\mathbf{D}$. In the online learning approach of [19], the only way to regularize $\mathbf{D}$ is to use a constraint set, on which we need to be able to project efficiently; this is unfortunately not always possible. In the matrix factorization framework of [13], it is argued that some applications can benefit from a structured penalty $\varphi$, but the approach of [13] is not easily amenable to stochastic optimization. Our approach makes it possible by using the proximal gradient surrogate

$$g_n : \mathbf{D} \mapsto \ell(\mathbf{x}_n, \mathbf{D}_{n-1}) + \mathrm{Tr}\left(\nabla_{\mathbf{D}}\ell(\mathbf{x}_n, \mathbf{D}_{n-1})^\top (\mathbf{D} - \mathbf{D}_{n-1})\right) + \frac{L}{2}\|\mathbf{D} - \mathbf{D}_{n-1}\|_F^2 + \varphi(\mathbf{D}). \quad (5)$$

It is indeed possible to show that $\mathbf{D} \mapsto \ell(\mathbf{x}_n, \mathbf{D})$ is differentiable, and its gradient is Lipschitz continuous with a constant $L$ that can be explicitly computed [18, 19].

We now design a proof-of-concept experiment. We consider a set of $N = 400\,000$ whitened natural image patches $\mathbf{x}_n$ of size $m = 20 \times 20$ pixels. We visualize some elements from a dictionary $\mathbf{D}$ trained by SPAMS [19] on the left of Figure 3; the dictionary elements are almost sparse, but have some residual noise among the small coefficients. Following [13], we propose to use a regularization function $\varphi$ encouraging neighbor pixels to be set to zero together, thus leading to a sparse structured dictionary. We consider the collection $\mathcal{G}$ of all groups of variables corresponding to squares of 4 neighbor pixels in $\{1, \ldots, m\}$. Then, we define $\varphi(\mathbf{D}) \triangleq \gamma_1 \sum_{j=1}^{K} \sum_{g \in \mathcal{G}} \max_{k \in g} |\mathbf{d}_j[k]| + \gamma_2\|\mathbf{D}\|_F^2$, where $\mathbf{d}_j$ is the $j$-th column of $\mathbf{D}$. The penalty $\varphi$ is a structured sparsity-inducing penalty that encourages groups of variables $g$ to be set to zero together [13]. Its proximal operator can be computed efficiently [20], and it is thus easy to use the surrogates (5). We set $\lambda_1 = 0.15$ and $\lambda_2 = 0.01$; after trying a few values for $\gamma_1$ and $\gamma_2$ at a reasonable computational cost, we obtain dictionaries with the desired regularization effect, as shown in Figure 3. Learning one dictionary of size $K = 256$ took a few minutes when performing one pass on the training data with mini-batches of size 100. This experiment demonstrates that our approach is more flexible and general than [13] and [19]. Note that it is possible to show that when $\gamma_2$ is large enough, the iterates $\mathbf{D}_n$ necessarily remain in a bounded set, and thus our convergence analysis presented in Section 3.3 applies to this experiment.



Figure 3: Left: Two visualizations of 25 elements from a larger dictionary obtained by the toolbox SPAMS [19]; the second view amplifies the small coefficients. Right: the corresponding views of the dictionary elements obtained by our approach after initialization with the dictionary on the left.

## 5 Conclusion

In this paper, we have introduced a stochastic majorization-minimization algorithm that gracefully scales to millions of training samples. We have shown that it has strong theoretical properties and some practical value in the context of machine learning. We have derived from our framework several new algorithms, which have shown to match or outperform the state of the art for solving large-scale convex problems, and to open up new possibilities for non-convex ones. In the future, we would like to study surrogate functions that can exploit the curvature of the objective function, which we believe is a crucial issue to deal with badly conditioned datasets.

**Acknowledgments**

This work was supported by the Gargantua project (program Mastodons - CNRS).

## Footnotes

[1]To alleviate this issue, it is possible to cut the dataset into $\eta$ mini-batches, reducing the memory load to $O(p\eta)$, which remains cumbersome when $p$ is very large.

[3] http://largescale.ml.tu-berlin.de.

[4] http://www.csie.ntu.edu.tw/~cjlin/libsvm/.

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
