[Supplementary Material]

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

## A Mathematical Background and Useful Results

In this paper, we use subdifferential calculus for convex functions. The definition of subgradients and directional derivatives can be found in classical textbooks, see, e.g., [2], [37]. We denote by $\partial f(\theta)$ the subdifferential of a convex function $f$ at a point $\theta$. Other definitions can be found in the appendix of [18], which uses a similar notation as ours.

In this section, we present several classical optimization and probabilistic tools, which we use in our paper. The first lemma is a classical quadratic upper-bound for differentiable functions with a Lipschitz gradient. It can be found for instance in Lemma 1.2.3 of [35], or in the appendix of [18].

**Lemma A.1** (**Convex Surrogate for Functions with Lipschitz Gradient**).
*Let $f : \mathbb{R}^p \to \mathbb{R}$ be differentiable and $\nabla f$ be $L$-Lipschitz continuous. Then, for all $\theta, \theta'$ in $\mathbb{R}^p$,*

$$|f(\theta') - f(\theta) - \nabla f(\theta)^\top (\theta' - \theta)| \leq \frac{L}{2} \|\theta - \theta'\|_2^2. \tag{6}$$

The next lemma is a simple relation, which will allow us to identify the subdifferential of a convex function with the one of its surrogate at a particular point.

**Lemma A.2** (**Surrogate Functions and Subdifferential**).
*Assume that $f, g : \mathbb{R}^p \to \mathbb{R}$ are convex, and that $h \triangleq g - f$ is differentiable at $\theta$ in $\mathbb{R}^p$ with $\nabla h(\theta) = 0$. Then, $\partial f(\theta) = \partial g(\theta)$.*

*Proof.* It is easy to show that $g$ and $f$ have the same directional derivatives at $\theta$ since $h$ is differentiable and $\nabla h(\theta) = 0$. This is sufficient to conclude that $\partial g(\theta) = \partial f(\theta)$ by using Proposition 3.1.6 of [2], a simple lemma relating directional derivatives and subgradients. $\qquad\square$

The following lemma is a lower bound for strongly convex functions. It can be found for instance in [36].

**Lemma A.3** (**Lower Bound for Strongly Convex Functions**).
*Let $f : \mathbb{R}^p \to \mathbb{R}$ be a $\mu$-strongly convex function. Let $z$ be in $\partial f(\kappa)$ for some $\kappa$ in $\mathbb{R}^p$. Then, the following inequality holds for all $\theta$ in $\mathbb{R}^p$:*

$$f(\theta) \geq f(\kappa) + z^\top (\theta - \kappa) + \frac{\mu}{2} \|\theta - \kappa\|_2^2.$$

*Proof.* The function $l : \theta \mapsto f(\theta) - \frac{\mu}{2} \|\theta - \kappa\|_2^2$ is convex by definition of strong convexity, and $l - f$ is differentiable with $\nabla(l - f)(\kappa) = 0$. We apply Lemma A.2, which tells us that $z$ is in $\partial l(\kappa)$. This is sufficient to conclude, by noticing that a convex function is always above its tangents. $\qquad\square$

The next lemma is also classical (see the appendix of [18]).

**Lemma A.4** (**Second-Order Growth Property**).
*Let $f : \mathbb{R}^p \to \mathbb{R}$ be a $\mu$-strongly convex function and $\Theta \subseteq \mathbb{R}^p$ be a convex set. Let $\theta^\star$ be the minimizer of $f$ on $\Theta$. Then, the following condition holds for all $\theta$ in $\Theta$:*

$$f(\theta) \geq f(\theta^\star) + \frac{\mu}{2} \|\theta - \theta^\star\|_2^2.$$

We now introduce a sequence of probabilistic tools, which we use in our convergence analysis for non-convex functions. The first one is a classical theorem on quasi-martingales, which was used in [3] for proving the convergence of the stochastic gradient descent algorithm.

**Theorem A.1** (**Convergence of Quasi-Martingales.**).
*This presentation follows [3] and Proposition 9.5 and Theorem 9.4 of [34]. The original theorem is due to [33]. Let $(\mathcal{F}_n)_{n \geq 0}$ be an increasing family of $\sigma$-fields. Let $(X_n)_{n \geq 0}$ be a real stochastic process such that every random variable $X_n$ is bounded below by a constant independent of $n$, and $\mathcal{F}_n$-measurable. Let*

$$\delta_n \triangleq \begin{cases} 1 & \text{if } \mathbb{E}[X_{n+1} - X_n | \mathcal{F}_n] > 0, \\ 0 & \text{otherwise.} \end{cases}$$

*If the series $\sum_{n=0}^{\infty} \mathbb{E}[\delta_n(X_{n+1} - X_n)]$ converges, then $(X_n)_{n\geq 0}$ is a quasi-martingale and converges almost surely to an integrable random variable $X_\infty$. Moreover,*

$$\sum_{n=0}^{\infty} \mathbb{E}\big[|\mathbb{E}[X_{n+1} - X_n | \mathcal{F}_n]|\big] < \infty.$$

The next lemma is simple, but useful to prove the convergence of deterministic algorithms.

### Lemma A.5. Deterministic Lemma on Non-negative Converging Series.
*Let $(a_n)_{n\geq 1}$, $(b_n)_{n\geq 1}$ be two non-negative real sequences such that the series $\sum_{n=1}^{\infty} a_n$ diverges, the series $\sum_{n=1}^{\infty} a_n b_n$ converges, and there exists $K > 0$ such that $|b_{n+1} - b_n| \leq K a_n$. Then, the sequence $(b_n)_{n\geq 1}$ converges to 0.*

*Proof.* The proof is inspired by the one of Proposition 1.2.4 of [31]. Since the series $\sum_{n\geq 1} a_n$ diverges, we necessarily have $\liminf_{n\to+\infty} b_n = 0$. Otherwise, it would be easy to contradict the assumption $\sum_{n\geq 1} a_n b_n < +\infty$.

Let us now proceed by contradiction and assume that $\limsup_{n\to+\infty} b_n = \lambda > 0$. We can then build two sequences of indices $(m_j)_{j\geq 1}$ and $(n_j)_{j\geq 1}$ such that

- $m_j < n_j < m_{j+1}$,

- $\frac{\lambda}{3} < b_k$, for $m_j \leq k < n_j$,

- $b_k \leq \frac{\lambda}{3}$, for $n_j \leq k < m_{j+1}$.

Let $\varepsilon = \frac{\lambda^2}{9K}$ and $\tilde{j}$ be large enough such that

$$\sum_{n=m_{\tilde{j}}}^{\infty} a_n b_n < \varepsilon.$$

Then, we have for all $j \geq \tilde{j}$ and all $m$ with $m_j \leq m \leq n_j - 1$,

$$|b_{n_j} - b_m| \leq \sum_{k=m}^{n_j-1} |b_{k+1} - b_k| \leq \frac{3K}{\lambda} \sum_{k=m}^{n_j-1} a_k \frac{\lambda}{3} \leq \frac{3K}{\lambda} \sum_{k=m}^{n_j-1} a_k b_k \leq \frac{3K}{\lambda} \sum_{k=m}^{+\infty} a_k b_k$$
$$\leq \frac{3K\varepsilon}{\lambda} \leq \frac{\lambda}{3}.$$

Therefore, by using the triangle inequality,

$$b_m \leq b_{n_j} + \frac{\lambda}{3} \leq \frac{2\lambda}{3}.$$

and finally, for all $m \geq \tilde{j}$,

$$b_m \leq \frac{2\lambda}{3},$$

which contradicts $\limsup_{n\to+\infty} b_n = \lambda > 0$. Therefore, $b_n \underset{n\to+\infty}{\longrightarrow} 0$.

$\square$

We now provide a stochastic version of Lemma A.6.

### Lemma A.6. Stochastic Lemma on Non-negative Converging Series.
*Let $(X_n)_{n\geq 1}$ be a sequence of non-negative measurable random variables on a probability space. Let also $a_n$, $b_n$ be two non-negative sequences such that $\sum_{n\geq 1} a_n = +\infty$ and $\sum_{n\geq 1} a_n b_n < +\infty$. Assume that there exists a constant $C$ such that for all $n \geq 1$, $\mathbb{E}[X_n] \leq b_n$ and $|X_{n+1} - X_n| \leq C a_n$ almost surely. Then $X_n$ almost surely converges to zero.*

*Proof.* The following series is convergent

$$\mathbb{E}\left[\sum_{n\geq 1} a_n X_n\right] = \sum_{n\geq 1}\mathbb{E}\left[a_n X_n\right] \leq \sum_{n\geq 1} a_n b_n < +\infty,$$

where we use the fact that the random variables are non-negative to interchange the sum and the expectation. We thus have that $\sum_{n\geq 1} a_n X_n$ converges with probability one. Then, let us call $a'_n = a_n$ and $b'_n = X_n$; the conditions of Lemma A.5 are satisfied for $a'_n$ and $b'_n$ with probability one, and $X_n$ almost surely converges to zero. $\square$

# B Auxiliary Lemmas

In this section, we present auxiliary lemmas for our convex and non-convex analyses. We start by presenting a lemma which is useful for both of them, and which is in fact a core component for all results presented in [18]. The proof of this lemma is simple and available in [18].

**Lemma B.1** (**Basic Properties of First-Order Surrogate Functions**).
*Let $g$ be in $\mathcal{S}_{L,\rho}(f,\kappa)$ for some $\kappa$ in $\Theta$. Define the approximation error function $h \triangleq g - f$ and let $\theta'$ be the minimizer of $g$ over $\Theta$. Then, for all $\theta$ in $\Theta$,*

- $\nabla h(\kappa) = 0$;

- $|h(\theta)| \leq \frac{L}{2}\|\theta - \kappa\|_2^2$;

- $f(\theta') \leq g(\theta') \leq f(\theta) + \frac{L}{2}\|\theta - \kappa\|_2^2 - \frac{\rho}{2}\|\theta - \theta'\|_2^2.$

## B.1 Convex Analysis

We introduce, for all $n \geq 0$, the quantity $\xi_n \triangleq \frac{1}{2}\mathbb{E}[\|\theta^\star - \theta_n\|_2^2]$, where $\theta^\star$ is a minimizer of $f$ on $\Theta$. Our analysis also involves several quantities that are defined recursively for all $n \geq 1$:

$$\begin{cases} A_n &\triangleq (1-w_n)A_{n-1} + w_n\xi_{n-1} \\ B_n &\triangleq (1-w_n)B_{n-1} + w_n\mathbb{E}[f(\theta_{n-1})] \\ C_n &\triangleq (1-w_n)C_{n-1} + \frac{(Rw_n)^2}{2\rho} \\ \bar{g}_n &\triangleq (1-w_n)\bar{g}_{n-1} + w_n g_n \\ \bar{f}_n &\triangleq (1-w_n)\bar{f}_{n-1} + w_n f_n \end{cases}, \tag{7}$$

where $A_0 \triangleq \frac{1}{L}(\rho\xi_0 - f^\star)$, $B_0 \triangleq 0$, $C_0 \triangleq 0$, $\bar{g}_0 = \bar{f}_0 \triangleq \theta \mapsto \frac{\rho}{2}\|\theta - \theta_0\|_2^2$. Note that $\bar{g}_0$ is $\rho$-strongly convex, and is minimized by $\theta_0$. The choice for $A_0, B_0, C_0$ is driven by technical reasons, which appear in the proof of Lemma B.4, a stochastic version of Lemma B.1.

Note that we also assume here that all the expectations above are well defined and finite-valued. In other words, we do not deal with measurability or integrability issues for simplicity, as often done in the literature [22].

**Lemma B.2** (**Auxiliary Lemma for Convex Analysis**).
*When the functions $f_n$ are convex, and the surrogates $g_n$ are in $\mathcal{S}_{L,\rho}(f_n, \theta_{n-1})$, we have under assumption (A) that for all $n \geq 1$,*

$$\bar{g}_n(\theta_{n-1}) \leq \bar{g}_n(\theta_n) + \frac{(Rw_n)^2}{2\rho}.$$

*Proof.* First, we remark that the subdifferentials of $g_n$ and $f_n$ at $\theta_{n-1}$ coincide by applying Lemma A.2. Then, we choose $z_n$ in $\partial g_n(\theta_{n-1}) = \partial f_n(\theta_{n-1})$, which is bounded by $R$ accord-

ing to assumption **(A)**, and we have

$$\bar{g}_n(\theta_n) = (1 - w_n)\bar{g}_{n-1}(\theta_n) + w_n g_n(\theta_n)$$

$$\geq (1-w_n)\left(\bar{g}_{n-1}(\theta_{n-1}) + \frac{\rho}{2}\|\theta_n - \theta_{n-1}\|_2^2\right) + w_n\left(g_n(\theta_{n-1}) + z_n^\top(\theta_n - \theta_{n-1}) + \frac{\rho}{2}\|\theta_n - \theta_{n-1}\|_2^2\right)$$

$$= \bar{g}_n(\theta_{n-1}) + w_n z_n^\top(\theta_n - \theta_{n-1}) + \frac{\rho}{2}\|\theta_n - \theta_{n-1}\|_2^2$$

$$\geq \bar{g}_n(\theta_{n-1}) - R w_n \|\theta_n - \theta_{n-1}\|_2 + \frac{\rho}{2}\|\theta_n - \theta_{n-1}\|_2^2$$

$$\geq \bar{g}_n(\theta_{n-1}) - \frac{(R w_n)^2}{2\rho}.$$

The first inequality uses Lemma A.4 and Lemma A.3 since $g_n$ is $\rho$-strongly convex by definition (and by induction $\bar{g}_n$ is $\rho$-strongly convex as well); the second inequality uses Cauchy-Schwarz's inequality and the fact that the subgradients of the functions $f_n$ are bounded by $R$. □

**Lemma B.3** (**Another Auxiliary Lemma for Convex Analysis**).
*When the functions $f_n$ are convex, and the surrogates $g_n$ are in $\mathcal{S}_{L,\rho}(f_n, \theta_{n-1})$, we have under assumption **(A)** that for all $n \geq 0$,*

$$B_n \leq \mathbb{E}[\bar{g}_n(\theta_n)] + C_n, \tag{8}$$

*Proof.* We proceed by induction, and start by showing that Eq. (8) is true for $n = 0$.

$$B_0 = 0 = \mathbb{E}[\bar{g}_0(\theta_0)] = \mathbb{E}[\bar{g}_0(\theta_0)] + C_0.$$

Let us now assume that it is true for $n - 1$, and show that it is true for $n$.

$$B_n = (1 - w_n)B_{n-1} + w_n\mathbb{E}[f(\theta_{n-1})]$$

$$\leq (1 - w_n)(\mathbb{E}[\bar{g}_{n-1}(\theta_{n-1})] + C_{n-1}) + w_n\mathbb{E}[f(\theta_{n-1})]$$

$$= (1 - w_n)(\mathbb{E}[\bar{g}_{n-1}(\theta_{n-1})] + C_{n-1}) + w_n\mathbb{E}[f_n(\theta_{n-1})]$$

$$= (1 - w_n)(\mathbb{E}[\bar{g}_{n-1}(\theta_{n-1})] + C_{n-1}) + w_n\mathbb{E}[g_n(\theta_{n-1})]$$

$$= \mathbb{E}[\bar{g}_n(\theta_{n-1})] + (1 - w_n)C_{n-1}$$

$$\leq \mathbb{E}[\bar{g}_n(\theta_n)] + \frac{(R w_n)^2}{2\rho} + (1 - w_n)C_{n-1}$$

$$= \mathbb{E}[\bar{g}_n(\theta_n)] + C_n.$$

The first inequality uses the induction hypothesis; the last inequality uses Lemma B.2 and the definition of $C_n$. We also used the fact that $\mathbb{E}[f_n(\theta_{n-1})] = \mathbb{E}[\mathbb{E}[f_n(\theta_{n-1})]|\mathcal{F}_{n-1}]] = \mathbb{E}[\mathbb{E}[f(\theta_{n-1})]|\mathcal{F}_{n-1}]] = \mathbb{E}[f(\theta_{n-1})]$, where $\mathcal{F}_{n-1}$ corresponds to the filtration induced by the past information before time $n$, such that $\theta_{n-1}$ is deterministic given $\mathcal{F}_{n-1}$. □

The next lemma is important; it is the stochastic version of Lemma B.1 for first-order surrogates.

**Lemma B.4** (**Basic Properties of Stochastic First-Order Surrogates**).
*When the functions $f_n$ are convex and the functions $g_n$ are in $\mathcal{S}_{L,\rho}(f_n, \theta_{n-1})$, we have under assumption **(A)** that for all $n \geq 0$,*

$$B_n \leq f^\star + L A_n - \rho\xi_n + C_n,$$

*Proof.* According to Lemma B.3, it is sufficient to show that $\mathbb{E}[\bar{g}_n(\theta_n)] \leq f^\star + L A_n - \rho\xi_n$ for all $n \geq 0$. Since $\bar{g}_n$ is $\rho$-strongly convex and $\theta_n$ is the minimizer of $\bar{g}_n$ over $\Theta$, we have $\mathbb{E}[\bar{g}_n(\theta_n)] \leq \mathbb{E}[\bar{g}_n(\theta^\star)] - \rho\xi_n$, by using Lemma A.4. Thus, it is in fact sufficient to show that $\mathbb{E}[\bar{g}_n(\theta^\star)] \leq f^\star + L A_n$. For $n = 0$, this inequality holds since $\mathbb{E}[\bar{g}_0(\theta^\star)] = \rho\xi_0 = f^\star + L A_0$. We can then proceed again by induction: assume that $\mathbb{E}[\bar{g}_{n-1}(\theta^\star)] \leq f^\star + L A_{n-1}$. Then,

$$\mathbb{E}[\bar{g}_n(\theta^\star)] = (1 - w_n)\mathbb{E}[\bar{g}_{n-1}(\theta^\star)] + w_n\mathbb{E}[g_n(\theta^\star)]$$

$$\leq (1 - w_n)(f^\star + L A_{n-1}) + w_n(\mathbb{E}[f_n(\theta^\star)] + L\xi_{n-1})$$

$$= (1 - w_n)(f^\star + L A_{n-1}) + w_n(f^\star + L\xi_{n-1})$$

$$= f^\star + L A_n,$$

where we have used Lemma B.1 to upper-bound the difference $\mathbb{E}[g_n(\theta^\star)] - \mathbb{E}[f_n(\theta^\star)]$ by $\xi_{n-1}$. □

For strongly-convex functions, we also have the following simple but useful relation between $A_n$ and $B_n$.

**Lemma B.5 (Relation between $A_n$ and $B_n$).**
*Under assumption (B), if $w_1 = 1$, we have for all $n \geq 1$,*

$$f^\star + \mu A_n \leq B_n.$$

*Proof.* This relation is true for $n = 1$ since we have $f^\star + \mu A_1 = f^\star + \mu \xi_0 \leq f(\theta_0) = B_1$ by applying Lemma A.4, since $f$ is $\mu$-strongly convex according to assumption (B). The rest follows by induction. $\square$

## B.2 Non-convex Analysis

When the functions $f_n$ are not convex, the convergence analysis becomes more involved. One key tool we use is a uniform convergence result when the function class $\{\mathbf{x} \mapsto \ell(\theta, \mathbf{x}) : \theta \in \Theta\}$ is "simple enough" in terms of entropy. Under the assumptions made in our paper, it is indeed possible to use some results from empirical processes [27], which provides us the following lemma.

**Lemma B.6 (Uniform Convergence).**
*Under assumptions (A), (C), and (D), we have the following uniform law of large numbers:*

$$\mathbb{E}\left[ \sup_{\theta \in \Theta} \left| \frac{1}{n} \sum_{i=1}^{n} f_i(\theta) - f(\theta) \right| \right] \leq \frac{C}{\sqrt{n}}, \tag{9}$$

*where $C$ is a constant, and $\sup_{\theta \in \Theta} \left| \frac{1}{n} \sum_{i=1}^{n} f_i(\theta) - f(\theta) \right|$ converges almost surely to zero.*

*Proof.* We simply refer to Lemma 19.36 and Example 19.7 of [27], where assumptions (C) and (D) ensure uniform boundness and squared integrability conditions. Note that we assume that the quantities $\sup_{\theta \in \Theta} \left| \frac{1}{n} \sum_{i=1}^{n} f_i(\theta) - f(\theta) \right|$ are measurable. This assumption does not incur a loss of generality, since measurability issues for empirical processes can be dealt with rigorously [27]. $\square$

The next lemma shows that uniform convergence applies to the weighted empirical risk $\bar{f}_n$, defined in Eq. (7), but with a different rate.

**Lemma B.7 (Uniform Convergence for $\bar{f}_n$).**
*Under assumptions (A), (C), (D), and (E), we have for all $n \geq 1$,*

$$\mathbb{E}\left[ \sup_{\theta \in \Theta} \left| \bar{f}_n(\theta) - f(\theta) \right| \right] \leq C w_n \sqrt{n},$$

*where $C$ is the same as in Lemma B.6, and $\sup_{\theta \in \Theta} \left| \bar{f}_n(\theta) - f(\theta) \right|$ converges almost surely to zero.*

*Proof.* We prove the two parts of the lemma separately. As in Lemma B.6, we assume all the quantities of interest to be measurable.

**First part of the lemma:**
Let us fix $n > 0$. It is easy to show that $\bar{f}_n$ can be written as $\bar{f}_n = \sum_{i=1}^{n} w_n^i f_i$ for some non-negative weights $w_n^i$ with $w_n^n = w_n$. Let us also define the empirical cost $F_i \triangleq \frac{1}{n-i+1} \sum_{j=i}^{n} f_j$. According to (9), we have $\mathbb{E}\left[ \sup_{\theta \in \Theta} |F_i(\theta) - f(\theta)| \right] \leq \frac{C}{\sqrt{n-i+1}}$. We now remark that

$$\bar{f}_n - f = \sum_{i=1}^{n} (w_n^i - w_n^{i-1})(n - i + 1)(F_i - f),$$

where we have defined $w_n^0 \triangleq 0$. This relation can be proved by simple calculation. We obtain the first part by using the triangle inequality, and the fact that $w_n^i \geq w_n^{i-1}$ for all $i$:

$$
\begin{aligned}
\mathbb{E}\left[\sup_{\theta \in \Theta} |\bar{f}_n(\theta) - f(\theta)|\right] &\leq \mathbb{E}\left[\sum_{i=1}^n (w_n^i - w_n^{i-1})(n - i + 1) \sup_{\theta \in \Theta} |F_i(\theta) - f(\theta)|\right] \\
&= \sum_{i=1}^n (w_n^i - w_n^{i-1})(n - i + 1) \mathbb{E}\left[\sup_{\theta \in \Theta} |F_i(\theta) - f(\theta)|\right] \\
&\leq \sum_{i=1}^n (w_n^i - w_n^{i-1}) C \sqrt{n - i + 1} \\
&\leq \sqrt{n} C \sum_{i=1}^n (w_n^i - w_n^{i-1}) \\
&= C\sqrt{n} w_n.
\end{aligned}
$$

This is unfortunately not sufficient to show that $\mathbb{E}\left[\sup_{\theta \in \Theta} |\bar{f}_n(\theta) - f(\theta)|\right]$ converges to zero almost surely. We will show this fact by using Lemma A.6.

**Second part of the lemma:**
We call $X_n = \sup_{\theta \in \Theta} |\bar{f}_n(\theta) - f(\theta)|$. We have

$$
\begin{aligned}
X_n - X_{n-1} &= \sup_{\theta \in \Theta} |(1 - w_n)(\bar{f}_{n-1}(\theta) - f(\theta)) + w_n(f_n(\theta) - f(\theta))| - X_{n-1} \\
&\leq \sup_{\theta \in \Theta} w_n |f_n(\theta) - f(\theta)| - w_n X_{n-1} \leq 2M w_n
\end{aligned}
$$

Let us denote by $\theta_n^\star$ a point in $\Theta$ such that $X_n = |\bar{f}_n(\theta_n^\star) - f(\theta_n^\star)|$. We also have

$$
\begin{aligned}
X_n - X_{n-1} &= \sup_{\theta \in \Theta} |(1 - w_n)(\bar{f}_{n-1}(\theta) - f(\theta)) + w_n(f_n(\theta) - f(\theta))| - X_{n-1} \\
&\geq (1 - w_n) X_{n-1} + w_n(f_n(\theta_{n-1}^\star) - f(\theta_{n-1}^\star)) - X_{n-1} \\
&\geq -w_n X_{n-1} + w_n(f_n(\theta_{n-1}^\star) - f(\theta_{n-1}^\star)) \\
&\geq -w_n 4M,
\end{aligned}
$$

where we use again the fact that all functions $f_n$, $\bar{f}_n$ and $f$ are bounded by $M$. Thus, we have shown that $|X_n - X_{n-1}| \leq 4M w_n$. Call $a_n = w_n$ and $b_n = w_n \sqrt{n}$, then the conditions of Lemma A.6 are satisfied, and $X_n$ converges almost surely to zero. $\qquad\square$

Finally, the next lemma illustrates why the strong convexity of the surrogates is important.

**Lemma B.8 (Stability of the Estimates).**
*When $g_n$ is in $\mathcal{S}_{L,\rho}(f, \theta_{n-1})$,*

$$
\|\theta_n - \theta_{n-1}\|_2 \leq \frac{2R w_n}{\rho}.
$$

*Proof.* Because the surrogates $g_n$ are $\rho$-strongly convex, we have from Lemma A.4

$$
\begin{aligned}
\frac{\rho}{2}\|\theta_n - \theta_{n-1}\|_2^2 &\leq \bar{g}_n(\theta_{n-1}) - \bar{g}_n(\theta_n) \\
&= w_n \left(g_n(\theta_{n-1}) - g_n(\theta_n)\right) + (1 - w_n) \left(\bar{g}_{n-1}(\theta_{n-1}) - \bar{g}_{n-1}(\theta_n)\right) \\
&\leq w_n \left(g_n(\theta_{n-1}) - g_n(\theta_n)\right) \\
&\leq w_n \left(f_n(\theta_{n-1}) - f_n(\theta_n)\right) \\
&\leq R w_n \|\theta_n - \theta_{n-1}\|_2.
\end{aligned}
$$

The second inequality comes from the fact that $\theta_{n-1}$ is a minimizer of $\bar{g}_{n-1}$; the third inequality is because $g_n(\theta_{n-1}) = f_n(\theta_{n-1})$ and $g_n \geq f_n$. This is sufficient to conclude. $\qquad\square$

# C Proofs of the Main Lemmas and Propositions

## C.1 Proof of Proposition 3.1

*Proof.* According to Lemma B.4, we have for all $n \geq 1$,

$$w_n B_{n-1} \leq w_n f^\star + L w_n A_{n-1} - L w_n \xi_{n-1} + w_n C_{n-1}.$$

By using the relations (7), this is equivalent to

$$B_{n-1} - B_n + w_n \mathbb{E}[f(\theta_{n-1})] \leq w_n f^\star + L(A_{n-1} - A_n) + C_{n-1} - C_n + \frac{(Rw_n)^2}{2L}.$$

By summing these inequalities between $1$ and $n$, we obtain

$$B_0 - B_n + \sum_{k=1}^{n} w_k \mathbb{E}[f(\theta_{k-1})] \leq \left( \sum_{k=1}^{n} w_k \right) f^\star + LA_0 - LA_n - C_n + \sum_{k=1}^{n} \frac{(Rw_k)^2}{2L}.$$

Note that we also have

$$B_n \leq f^\star + LA_n + C_n = LA_n + C_n + B_0 - LA_0 + L\xi_0.$$

Therefore, by combining the two previous inequalities,

$$\sum_{k=1}^{n} w_k \mathbb{E}[f(\theta_{k-1})] \leq \left( \sum_{k=1}^{n} w_k \right) f^\star + L\xi_0 + \sum_{k=1}^{n} \frac{(Rw_k)^2}{2L},$$

and by using Jensen's inequality,

$$\mathbb{E}[f(\bar{\theta}_{n-1})] - f^\star \leq \frac{L\xi_0 + \frac{R^2}{2L} \sum_{k=1}^{n} w_k^2}{\sum_{k=1}^{n} w_k}.$$

$\square$

## C.2 Proof of Corollary 3.1

*Proof.*
We choose weights of the form $w_n \triangleq \frac{\gamma}{\sqrt{n}}$. Then, we have

$$\sum_{k=1}^{n} w_k^2 \leq \gamma^2 (1 + \log n),$$

by using the fact that $\sum_{k=1}^{n} \frac{1}{k} \leq 1 + \log(n)$. We also have for $n \geq 2$,

$$\sum_{k=1}^{n} w_k \geq 2\gamma(\sqrt{n+1} - 1) \geq \gamma\sqrt{n},$$

where we use the fact that $\sum_{k=1}^{n} \frac{1}{\sqrt{k}} \geq 2(\sqrt{n+1} - 1)$, and the fact that $2(\sqrt{n+1} - 1) \geq \sqrt{n}$ for all $n \geq 2$. Plugging this inequalities into (3) yields the desired result. $\square$

## C.3 Proof of Proposition 3.2

*Proof.* We proceed in several steps, proving the convergence rates of several quantities of interest.

**Convergence rate of $C_n$:**
Let us show by induction that we have $C_n \leq \frac{R^2}{\rho} w_n$ for all $n \geq 1$. This is obviously true for $n = 1$

by definitions of $w_1 = 1$ and $C_1 = \frac{R^2}{2\rho}$. Let us now assume that it is true for $n-1$. We have

$$
\begin{aligned}
C_n &= (1 - w_n)C_{n-1} + \frac{R^2}{2\rho}w_n^2 \\
&\leq \frac{R^2}{\rho}w_n\left((1 - w_n)\frac{w_{n-1}}{w_n} + \frac{w_n}{2}\right) \\
&\leq \frac{R^2}{\rho}w_n\left(\frac{\beta(n-1)}{\beta n + 1}\frac{\beta n + 1}{\beta(n-1) + 1} + \frac{1}{\beta n + 1}\right) \\
&\leq \frac{R^2}{\rho}w_n\left(\frac{\beta(n-1)}{\beta(n-1) + 1} + \frac{1}{\beta(n-1) + 1}\right) \\
&= \frac{R^2}{\rho}w_n.
\end{aligned}
\tag{10}
$$

We conclude by induction that this is true for all $n \geq 1$.

**Convergence rate of $A_n$:**
From Lemma B.5 and B.4, we have for all $n \geq 2$,

$$\mu A_{n-1} \leq L A_{n-1} - \rho \xi_{n-1} + C_{n-1}.$$

Multiplying this inequality by $w_n$,

$$2\mu w_n A_{n-1} \leq \rho w_n (A_{n-1} - \xi_{n-1}) + w_n C_{n-1},$$

where the factor 2 comes from the fact that $\rho = L + \mu$. By using the definition of $A_n$ in Eq. (7), we obtain the relation

$$A_n \leq \left(1 - \frac{2\mu w_n}{\rho}\right)A_{n-1} + \frac{w_n}{\rho}C_{n-1}.$$

Let us now show by induction that we have, for all $n \geq 1$, the convergence rate $A_n \leq \delta w_n$, where $\delta \triangleq \max\left(\frac{R^2}{\rho\mu}, \xi_0\right)$. For $n = 1$, we have that and $w_1 = 1$, and thus $A_1 = \xi_0 \leq \delta$. Assume now that we have $A_{n-1} \leq \delta w_{n-1}$ for some $n \geq 1$. Then, by using the convergence rate (10) and the induction hypothesis,

$$
\begin{aligned}
A_n &\leq \delta w_n\left(\left(1 - \frac{2\mu w_n}{\rho}\right)\frac{w_{n-1}}{w_n} + \frac{R^2 w_{n-1}}{\rho^2 \delta}\right) \\
&\leq \delta w_n\left(\left(1 - \frac{2\mu w_n}{\rho}\right)\frac{w_{n-1}}{w_n} + \mu\frac{w_{n-1}}{\rho}\right) \\
&\leq \delta w_n\left(\frac{\beta n + 1 - \frac{2\mu(1+\beta)}{\rho}}{\beta n + 1}\frac{\beta n + 1}{\beta(n-1) + 1} + \frac{\frac{\mu(1+\beta)}{\rho}}{\beta(n-1) + 1}\right) \\
&= \delta w_n\left(\frac{\beta n + 1 - \frac{\mu(1+\beta)}{\rho}}{\beta(n-1) + 1}\right) \\
&\leq \delta w_n.
\end{aligned}
$$

The last inequality uses the fact that $\frac{\mu(1+\beta)}{\rho} \geq \beta$ because $\beta \leq \frac{\mu}{L}$. we conclude by induction that $A_n \leq \delta w_n$ for all $n \geq 1$.

**Convergence rate of $\mathbb{E}[f(\hat\theta_n) - f^\star] + \rho\xi_n$:**
We use again Lemma B.4:

$$B_n - f^\star + \rho\xi_n \leq L A_n + C_n,$$

and we consider two possible cases

- If $\frac{R^2}{\rho\mu} \geq \xi_0$, then

$$B_n - f^\star + \rho\xi_n \leq \frac{R^2}{\rho}\left(1 + \frac{L}{\mu}\right)w_n$$
$$= \frac{R^2}{\mu}w_n$$
$$\leq \frac{2R^2}{\mu(\beta n + 1)},$$

where we simply use the convergence rates of $A_n$ and $C_n$ computed before.

- If instead $\frac{R^2}{\rho\mu} < \xi_0$, then

$$B_n - f^\star + \rho\xi_n \leq \left(\frac{R^2}{\rho} + L\xi_0\right)w_n$$
$$\leq \rho\xi_0 w_n$$
$$\leq \frac{2\rho\xi_0}{\beta n + 1}.$$

It is then easy to prove that $\mathbb{E}[f(\hat{\theta}_n) - f^\star] \leq B_n$ by using Jensen's inequality, which allows us to conclude.

$\square$

## C.4  Proof of Proposition 3.3

*Proof.* We generalize the proof of convergence for online matrix factorization of [19]. The proof exploits Theorem A.1 about the convergence of quasi-martingales [33], similarly as [3] for proving the convergence of the stochastic gradient descent algorithm for non-convex functions.

**Almost sure convergence of** $(\bar{g}_n(\theta_n))_{n\geq 1}$**:**
The first step consists of applying a convergence theorem for the sequence $(\bar{g}_n(\theta_n))_{n\geq 1}$ by bounding its positive expected variations. Define $Y_n \triangleq \bar{g}_n(\theta_n)$. For $n \geq 2$, we have

$$
\begin{aligned}
Y_n - Y_{n-1} &= \bar{g}_n(\theta_n) - \bar{g}_n(\theta_{n-1}) + \bar{g}_n(\theta_{n-1}) - \bar{g}_{n-1}(\theta_{n-1}) \\
&= (\bar{g}_n(\theta_n) - \bar{g}_n(\theta_{n-1})) + w_n(g_n(\theta_{n-1}) - \bar{g}_{n-1}(\theta_{n-1})) \\
&= (\bar{g}_n(\theta_n) - \bar{g}_n(\theta_{n-1})) + w_n(\bar{f}_{n-1}(\theta_{n-1}) - \bar{g}_{n-1}(\theta_{n-1})) + w_n(g_n(\theta_{n-1}) - \bar{f}_{n-1}(\theta_{n-1})) \\
&= (\bar{g}_n(\theta_n) - \bar{g}_n(\theta_{n-1})) + w_n(\bar{f}_{n-1}(\theta_{n-1}) - \bar{g}_{n-1}(\theta_{n-1})) + w_n(f_n(\theta_{n-1}) - \bar{f}_{n-1}(\theta_{n-1})) \\
&\leq w_n(f_n(\theta_{n-1}) - \bar{f}_{n-1}(\theta_{n-1})).
\end{aligned}
$$
(11)

The final inequality comes from the inequality $\bar{g}_n \geq \bar{f}_n$, which is easy to show by induction starting from $n = 1$ since $w_1 = 1$. It follows,

$$
\begin{aligned}
\mathbb{E}[\bar{g}_n(\theta_n) - \bar{g}_{n-1}(\theta_{n-1})|\mathcal{F}_{n-1}] &\leq w_n \mathbb{E}[f_n(\theta_{n-1}) - \bar{f}_{n-1}(\theta_{n-1})|\mathcal{F}_{n-1}] \\
&= w_n(f(\theta_{n-1}) - \bar{f}_{n-1}(\theta_{n-1})) \\
&\leq w_n \sup_{\theta \in \Theta} |f(\theta) - \bar{f}_{n-1}(\theta)|,
\end{aligned}
$$

where $\mathcal{F}_{n-1}$ is the filtration representing the past information before time $n$. Call now

$$
\delta_n \triangleq \begin{cases} 1 & \text{if } \mathbb{E}[\bar{g}_n(\theta_n) - \bar{g}_{n-1}(\theta_{n-1})|\mathcal{F}_{n-1}] > 0 \\ 0 & \text{otherwise.} \end{cases}
$$

Then, the series below with non-negative summands converges:

$$\sum_{n=1}^{\infty} \mathbb{E}[\delta_n(\bar{g}_n(\theta_n) - \bar{g}_{n-1}(\theta_{n-1}))] = \sum_{n=1}^{\infty} \mathbb{E}[\delta_n \mathbb{E}[(\bar{g}_n(\theta_n) - \bar{g}_{n-1}(\theta_{n-1}))|\mathcal{F}_{n-1}]]$$

$$\leq \sum_{n=1}^{\infty} \mathbb{E}\left[w_n \sup_{\theta \in \Theta} |f(\theta) - \bar{f}_{n-1}(\theta)|\right]$$

$$\leq \sum_{n=1}^{\infty} C w_n^2 \sqrt{n} < +\infty,$$

The second inequality comes from Lemma B.7. Since in addition $\bar{g}_n$ is bounded below by some constant independent of $n$, we can apply Theorem A.1. This theorem tells us that $(\bar{g}_n(\theta_n))_{n\geq 1}$ converges almost surely to an integrable random variable $g^\star$ and that $\sum_{n=1}^{\infty} \mathbb{E}[|\mathbb{E}[\bar{g}_n(\theta_n) - \bar{g}_{n-1}(\theta_{n-1})|\mathcal{F}_{n-1}]|]$ converges almost surely.

**Almost sure convergence of** $(\bar{f}_n(\theta_n))_{n\geq 1}$**:**
We will show by using Lemma A.5 that the non-positive term $\bar{f}_n(\theta_n) - \bar{g}_n(\theta_n)$ almost surely converges to zero, and thus $(\bar{f}_n(\theta_n))_{n\geq 1}$ is also converging almost surely to $g^\star$.

We observe that

$$\sum_{n=1}^{\infty} \mathbb{E}[|\mathbb{E}[\bar{g}_n(\theta_n) - \bar{g}_{n-1}(\theta_{n-1})|\mathcal{F}_{n-1}]|] = \mathbb{E}\left[\sum_{n=1}^{\infty} |\mathbb{E}[\bar{g}_n(\theta_n) - \bar{g}_{n-1}(\theta_{n-1})|\mathcal{F}_{n-1}]|\right] < +\infty.$$

Thus, the series $\sum_{n=1}^{\infty} |\mathbb{E}[\bar{g}_n(\theta_n) - \bar{g}_{n-1}(\theta_{n-1})|\mathcal{F}_{n-1}]|$ is absolutely convergent with probability one, and the series $\sum_{n=1}^{\infty} \mathbb{E}[\bar{g}_n(\theta_n) - \bar{g}_{n-1}(\theta_{n-1})|\mathcal{F}_{n-1}]$ is also almost surely convergent.

We also remark that, using Lemma B.7,

$$\mathbb{E}\left[\sum_{n=1}^{+\infty} w_n|f(\theta_{n-1}) - \bar{f}_{n-1}(\theta_{n-1})|\right] \leq C \sum_{n=1}^{+\infty} w_n^2 \sqrt{n} < +\infty,$$

and thus $w_n(f(\theta_{n-1}) - \bar{f}_{n-1}(\theta_{n-1}))$ is the summand of an absolutely convergent series with probability one.

Taking the expectation of Eq. (11) conditioned on $\mathcal{F}_{n-1}$, it remains that the non-positive term $w_n(\bar{f}_{n-1}(\theta_{n-1}) - \bar{g}_{n-1}(\theta_{n-1}))$ is also necessarily the summand of an almost surely convergent series, since all other terms in the equation are summands of almost surely converging sums. This is not sufficient to immediately conclude that $\bar{f}_n(\theta_n) - \bar{g}_n(\theta_n)$ converges to zero almost surely, and thus we will use Lemma A.5. We have that $\sum_{n=1}^{+\infty} w_n$ diverges, that $\sum_{n=1}^{+\infty} w_n(\bar{g}_{n-1}(\theta_{n-1}) - \bar{f}_{n-1}(\theta_{n-1}))$ converges almost surely. Define $X_n \triangleq (\bar{g}_{n-1}(\theta_{n-1}) - \bar{f}_{n-1}(\theta_{n-1}))$. By definition of the surrogate functions, the differences $h_n \triangleq g_n - f_n$ are differentiable and their gradients are $L$-Lipschitz continuous. Since in addition $\Theta$ is compact and $\nabla h_n(\theta_{n-1}) = 0$, $\nabla h_n$ is bounded by some constant $R'$ independent of $n$, and the function $h_n$ is $R'$-Lipschitz. This is therefore also the case for $\bar{h}_n = \bar{g}_n - \bar{f}_n$.

$$|X_{n+1} - X_n| = |\bar{h}_n(\theta_n) - \bar{h}_{n-1}(\theta_{n-1})|$$

$$\leq |\bar{h}_n(\theta_n) - \bar{h}_n(\theta_{n-1})| + |\bar{h}_n(\theta_{n-1}) - \bar{h}_{n-1}(\theta_{n-1})|$$

$$\leq R'\|\theta_n - \theta_{n-1}\|_2 + |\bar{h}_n(\theta_{n-1}) - \bar{h}_{n-1}(\theta_{n-1})|$$

$$\leq \frac{2RR'}{\rho} w_n + w_n|h_n(\theta_{n-1}) - \bar{h}_{n-1}(\theta_{n-1})|$$

$$= \frac{2RR'}{\rho} w_n + w_n|\bar{h}_{n-1}(\theta_{n-1})|$$

$$\leq O(w_n).$$

The second inequality uses the fact that $\bar{h}_n$ is $R'$-Lipschitz; The second inequality uses Lemma B.8; the last equality uses the fact that the functions $h_n$ are also bounded by some constant independent of $n$ (using the fact that $\nabla h_n$ is uniformly bounded). We can now apply Lemma A.5, and $X_n$ converges to zero with probability one. Thus, $(\bar{f}_n(\theta_n))_{n\geq 1}$ converges almost surely to $g^\star$.

**Almost sure convergence of** $(f(\theta_n))_{n \geq 1}$**:**
Since $(\bar{f}_n(\theta_n))_{n \geq 1}$ converges almost surely, we simply use Lemma A.6, which tells us that $\bar{f}_n$ converges uniformly to $f$. Then, $(f(\theta_n))_{n \geq 1}$ converges almost surely to $g^\star$.

**Asymptotic Stationary Point Condition:**
Let us call $\bar{h}_n \triangleq \bar{g}_n - \bar{f}_n$, which can be shown to be differentiable with a $L$-Lipschitz gradient by definition of the surrogate $g_n$. For all $\theta$ in $\Theta$,

$$\nabla \bar{f}_n(\theta_n, \theta - \theta_n) = \nabla \bar{g}_n(\theta_n, \theta - \theta_n) - \nabla \bar{h}_n(\theta_n)^\top (\theta - \theta_n).$$

Since $\theta_n$ is the minimizer of $\bar{g}_n$, we have $\nabla \bar{g}_n(\theta_n, \theta - \theta_n) \geq 0$.

Since $\bar{h}_n$ is differentiable and its gradient is $L$-Lipschitz continuous, we can apply Lemma A.1 to $\theta = \theta_n$ and $\theta' = \theta_n - \frac{1}{L}\nabla \bar{h}_n(\theta_n)$, which gives $\bar{h}_n(\theta') \leq \bar{h}_n(\theta_n) - \frac{1}{2L}\|\nabla \bar{h}_n(\theta_n)\|_2^2$. Since we have shown that $\bar{h}_n(\theta_n) = \bar{g}_n(\theta_n) - \bar{f}(\theta_n)$ converges to zero and $\bar{h}_n(\theta') \geq 0$, we have that $\|\nabla \bar{h}_n(\theta_n)\|_2$ converges to zero. Thus,

$$\inf_{\theta \in \Theta} \frac{\nabla \bar{f}_n(\theta_n, \theta - \theta_n)}{\|\theta - \theta_n\|_2} \geq -\|\nabla \bar{h}_n(\theta_n)\|_2 \xrightarrow[n \to +\infty]{} 0 \text{ a.s.}$$

$\square$

## C.5   Proof of Proposition 3.4

*Proof.* Since $\Theta$ is compact according to assumption **(C)**, the sequence $(\theta_n)_{n \geq 1}$ admits limit points. Let us consider a converging subsequence $(n_k)_{k \geq 1}$ to a limit point $\theta_\infty$ in $\Theta$. In this converging subsequence, we can also find a subsequence $(n_{k'})_{k' \geq 1}$ such that $\kappa_{n_{k'}}$ converges to a point $\kappa_\infty$ in $\mathcal{K}$ (which is compact). For the sake of simplicity, and without loss of generality, we remove the indices $k$ and $k'$ from the notation and assume that $\theta_n$ converges to $\theta_\infty$, while $\kappa_n$ converges to $\kappa_\infty$. It is then easy to see that the functions $\bar{g}_n$ converge uniformly to $\bar{g}_\infty \triangleq g_{\kappa_\infty}$, given the assumptions made in the proposition.

Defining $\bar{h}_\infty \triangleq \bar{g}_\infty - f$, we have for all $\theta$ in $\Theta$:

$$\nabla f(\theta_\infty, \theta - \theta_\infty) = \nabla \bar{g}_\infty(\theta_\infty, \theta - \theta_\infty) - \nabla \bar{h}_\infty(\theta_\infty, \theta - \theta_\infty).$$

To prove the proposition, we will first show that $\nabla \bar{g}_\infty(\theta_\infty, \theta - \theta_\infty) \geq 0$ and then that $\nabla \bar{h}_\infty(\theta_\infty, \theta - \theta_\infty) = 0$.

**Proof of** $\nabla \bar{g}_\infty(\theta_\infty, \theta - \theta_\infty) \geq 0$**:**
It is sufficient to show that $\theta_\infty$ is a minimizer of $\bar{g}_\infty$. This is straightforward, by taking the limit when $n$ goes to infinity of

$$\bar{g}_n(\theta) \geq \bar{g}_n(\theta_n),$$

where we use the uniform convergence of $\bar{g}_n$.

**Proof of** $\nabla \bar{h}_\infty(\theta_\infty, \theta - \theta_\infty) = 0$**:**
Since both $\bar{f}_n$ and $\bar{g}_n$ converges uniformly (according to Lemma B.7 for $\bar{f}_n$), we have that $\bar{h}_n$ converges uniformly to $\bar{h}_\infty$. Since $\bar{h}_n$ is differentiable with a $L$-Lipschitz gradient, we have for all vector $\mathbf{z}$ in $\mathbb{R}^p$,

$$\bar{h}_n(\theta_n + \mathbf{z}) = \bar{h}_n(\theta_n) + \nabla \bar{h}_n(\theta_n)^\top \mathbf{z} + O(\|\mathbf{z}\|_2^2),$$

where the constant in $O$ is independent of $n$. By taking the limit when $n$ goes to infinity, it remains

$$\bar{h}_\infty(\theta_\infty + \mathbf{z}) = \bar{h}_\infty(\theta_\infty) + O(\|\mathbf{z}\|_2^2),$$

since we have shown in the proof of Proposition 3.3 that $\|\nabla \bar{h}_n(\theta_n)\|_2$ converges to zero. Since $\bar{h}_\infty$ admits a first order extension around $\theta_\infty$ it is differentiable at this point and furthermore, $\nabla \bar{h}_\infty(\theta_\infty) = 0$. This is sufficient to conclude. $\square$

## C.6   Proof of Proposition 3.5

*Proof.* First we notice that

- $g_n \geq f_n$;

- $g_n(\theta_{n-1}) = f_n(\theta_{n-1})$;

- $g_n$ is $\rho_1$-strongly convex since $\theta \mapsto g_{k,n}(\gamma_k(\theta))$ can be shown to be convex, following elementary composition rules for convex functions (see [32], Section 3.2.4).

Thus, the only property missing is the smoothness of the approximation error $h_n \triangleq g_n - f_n$. Rather than writing again a full proof, we now simply review the different places where this property is used, and which modifications should be made to the proofs of Propositions 3.3 and 3.4.

In the second step of this proof, we require the functions $h_n$ to be uniformly Lipschitz and uniformly bounded. It easy to check that it is still the case with the assumptions we made in Proposition 3.5.

The last step about the asymptotic point condition is however more problematic, where we cannot show anymore that the quantity $\nabla \bar{h}_n(\theta_n)$ converges to zero (since $\bar{h}_n$ is not differentiable anymore). Instead, we need to show that the directional derivative $\frac{\nabla \bar{h}_n(\theta_n, \theta - \theta_n)}{\|\theta - \theta_n\|}$ uniformly converges to zero on $\Theta$.

We will show the result for $K = 1$; it will be easy to extend it to any arbitrary $K > 2$. We remark that

$$\nabla \bar{h}_n(\theta_n, \theta - \theta_n) = \nabla \bar{h}_{0,n}(\theta_n)^\top (\theta - \theta_n) + \lim_{t \to 0^+} \frac{\bar{h}_{1,n}(\gamma_1(\theta_n + t(\theta - \theta_n))) - \bar{h}_{1,n}(\gamma_1(\theta_n))}{t},$$

where $\bar{h}_{0,n}$ and $\bar{h}_{1,n}$ are defined similarly as $\bar{h}_n$ for the functions $h_{0,n} \triangleq g_{0,n} - f_{0,n}$ and $h_{1,n} \triangleq g_{1,n} - f_{1,n}$ respectively. Since $\bar{h}_n(\theta_n)$ is shown to converge to zero, we have that the non-negative quantities $\bar{h}_{0,n}(\theta_n)$ and $\bar{h}_{1,n}(\gamma_1(\theta_n))$ converge to zero as well. Since $\bar{h}_{0,n}$ and $\bar{h}_{1,n}$ are differentiable and their gradients are Lipschitz, we use similar arguments as in the proof of Proposition 3.3, and we have that $\nabla \bar{h}_{0,n}(\theta_n)$ and $\bar{h}'_{1,n}(\gamma_1(\theta_n))$ converge to zero (where $\bar{h}'_{1,n}$ is the derivative of $\bar{h}_{1,n}$. Concerning the second term, we can make the following Taylor expansion for $\bar{h}_{1,n}$:

$$\bar{h}_{1,n}(\gamma_1(\theta_n + \mathbf{z})) = \bar{h}_{1,n}(\gamma_1(\theta_n)) + \bar{h}'_{1,n}(\gamma_1(\theta_n))(\gamma_1(\theta_n + \mathbf{z}) - \gamma_1(\theta_n)) + O\left((\gamma_1(\theta_n + \mathbf{z}) - \gamma_1(\theta_n))^2\right),$$

where the constant in the $O$ notation is independent of $\theta_n$ and $\mathbf{z}$ (since the derivative is $L_1$-Lipschitz). Plugging $\mathbf{z} \triangleq t(\theta - \theta_n)$ in this last equation, and using the Lipschitz property of $\gamma_1$, we have

$$\lim_{t \to 0^+} \left| \frac{\bar{h}_{1,n}(\gamma_1(\theta_n + t(\theta - \theta_n))) - \bar{h}_{1,n}(\gamma_1(\theta_n))}{t} \right| \leq |\bar{h}'_{1,n}(\gamma_1(\theta_n))| \|\theta - \theta_n\|.$$

Since $\bar{h}'_{1,n}(\gamma_1(\theta_n))$ converges to zero, we can conclude the proof of the modified Proposition 3.3.

The proof of Proposition 3.4 can be modified with similar arguments. $\qquad \square$

## D   Additional Experimental Results

We present in Figures 4 and 5 some additional experimental comparisons, which complement the ones of Section 4.1. Figures 6 and 7 present additional plots from the experiment of Section 4.2. Finally, we present three dictionaries corresponding to the experiment of Section 4.3 in Figures 8, 9 and 10.

## Supplementary References

[31] D.P. Bertsekas. *Nonlinear programming*. Athena Scientific Belmont, 1999. 2nd edition.

[32] S.P. Boyd and L. Vandenberghe. *Convex Optimization*. Cambridge University Press, 2004.

[33] D. L. Fisk. Quasi-martingales. *T. Am. Math. Soc.*, 120(3):359–388, 1965.

[34] M. Métivier. *Semi-martingales*. Walter de Gruyter, 1983.

[35] Y. Nesterov. *Introductory lectures on convex optimization*. Kluwer Academic Publishers, 2004.

[36] Y. Nesterov and J.-P. Vial.   Confidence level solutions for stochastic programming.   *Automatica*, 44(6):1559–1568, 2008.

[37] J. Nocedal and S.J. Wright. *Numerical optimization*. Springer Verlag, 2006. 2nd edition.

Figure 4: Comparison between LIBLINEAR and SMM in the high regularization regime for $\ell_1$-logistic regression.

Figure 5: Comparison between LIBLINEAR and SMM in the low regularization regime for $\ell_1$-logistic regression.

Figure 6: Comparison between batch and online DC programming, with high regularization for the datasets rcv1 and webspam. Note that each iteration in the batch setting can perform several epochs.

Figure 7: Comparison between batch and online DC programming, with low regularization for the datasets rcv1 and webspam. Note that each iteration in the batch setting can perform several epochs.

Figure 8: Dictionary obtained using the toolbox SPAMS [19].

Figure 9: Sparse dictionary obtained by our approach, using the dictionary of Figure 8 as an initialization.

Figure 10: Sparse dictionary obtained by our approach, using the dictionary of Figure 8 as an initialization, and with a higher regularization parameter than in Figure 9.