[Reviews · NeurIPS 2013]

Submitted by Assigned_Reviewer_5

This paper proposes stochastic versions of majorization-minimization (MM) schemes for optimization (convex or non-convex), derives convergence rates for several scenarios, and reports experimental results on large-scale problems. The paper is well written, in a very precise and clear way, and it contains significant and high-quality work. In more than one sense, this paper builds upon and extends the work of [19], but this does not diminish its relevance, since the stochastic version of MM (as the authors correctly claim) requires a significantly more involved analysis.
Summary: As stated above, I believe this is a high-quality piece of work, with a relevant contribution to large-scale optimization methods for machine learning. I definitely recommend acceptance.

Submitted by Assigned_Reviewer_6

Majorization-minimization methods often arise in machine learning; the EM algorithm and the proximal gradient method are just two examples. The purpose of this paper is to present a stochastic version of this scheme that can handle large-scale problems. Overall, this is an interesting paper and the experimental results, especially on sparse logistic regression, are encouraging.

Some specific comments:

(*) I might suggest changing the title of the paper to make it more inviting to a wider audience [note: published version has been changed from the original].

(*) On the top of page 3, it is worth fleshing out exactly what a proximal gradient step is and explicitly giving the expression prox_{f_2}(...) that is obtained.

(*) Algorithm 1 is clearly presented and easy to understand, but it is worth making a bit clearer up front in the text what the three options are referring to.

(*) Some more motivation on the regularizer mentioned in 4.2 would help.

(*) Some additional discussion would help in section 3 to motivate the design of Algorithm 1. In particular, some (even brief) commentary on how to think about the approximate surrogate, its the weighted updates for each example, and the different averaging schemes for the weights would be worthwhile.
Summary: This is an interesting paper, but a few details should be fleshed out more, and overall, it would be helpful if the authors made some more effort to make the paper a bit more inviting and readable for a wider audience.

Submitted by Assigned_Reviewer_7

Summary:
This paper suggests a stochastic algorithm based on majorization-minimization principle, i.e., iteratively minimizing a majorizing surrogate of an objective function, for convex, strongly convex and also certain non-convex (DC) optimization problems. The majorization-minimization principle is known to underlie a number of known algorithms such as expectation-maximization (EM). The theoretical results are also supported with various numerical experiments.

Quality:
Overall the problem is well-motivated and the conditions used in the analyses, theoretical developments and their proofs (I did not check the ones used for the math background) seem correct. Yet the paper is missing a number of quite related literature on first order methods for nonconvex problems such as “Stochastic first-and zeroth-order methods for nonconvex stochastic programming“ by Ghadimi and Lan. Also the logarithmic loss in Corollary 3.2 due to decreasing step sizes indicates this method is not an optimal algorithm. The same is true for strongly convex function case given in Proposition 3.2. This should perhaps be emphasized a bit clearer.

The paper seems to be very closely connected with the reference [19]. The authors should state for each proposition, lemma, etc., whether they were from [19] or not, or whether the corresponding analysis from [19] easily extends to this case or not.

The numerical results seem promising. Yet there is a caveat between the theory presented and numerical results. First of all, almost all of the theory is built upon assuming that the domain of the optimization problem to be solved is bounded, i.e., for the stepsizes theoretically analyzed one needs to know an upper bound on \|\theta^*-\theta_0\|_2 where \theta^* is t he optimum solution and \theta_0 is the initial point. Also this quantity naturally occurs, as is the case with other first order methods, in the convergence rates. Yet in the experiments, all of the problems tested have unbounded domains. Moreover, the convergence analysis of the algorithms are carried over for specific stepsize rules, i.e., {\gamma\over \sqrt{n}} etc. On the other hand, the step size rules used in the experiments have quite different forms, and changes from problem to problem. I think this is a bit disturbing. The authors should either give justification for the step sizes used in the experiments theoretically or should use the ones they have build the theory for (at least for comparison purposes, they should be included in the experiments, anyway). Comparisons against FISTA is nice yet there are better (optimal) algorithms for strongly convex functions, and comparisons against those should also be presented.

Clarity:
The paper is mostly written in a clear fashion, with a reasonable logical structure. On the other hand, reading the supplemental material is almost essential in terms of digesting the notation, etc. I guess given the page limitation, there is no other way to fix this issue either. There are a few minor typos, grammatical errors. Also certain terms and notation such as DC or use of options 2 & 3 in Algorithm 1, $\overline{f}_n(\cdot)$ should be introduced earlier.

Originality & Significance:
Despite the connections with the existing algorithms, to the best of my knowledge the results seem original and valuable (especially in the nonconvex case).

Minor issues:
a. On page 4, line 180: “For space limitation details, all proofs…” should be “For space limitation all proofs…”
b. On page 4, line 198: \log(\sqrt(n) should be \log(n)
c. On page 5, before Proposition 3.5, stating the motivation for the special partially separable structure considered in Proposition 3.5 will be nicer.
d. On page 6, lines 284-285, description of the step size rule is confusing. How do you select w_n?
e. On page 6, how did you decide on the sizes of the training and test sets for the datasets?
f. On page 7, line 234 define what an ``epoch” is
g. On page 10, line 491: “notation as us” should be “notation to us”
h. On page 14, line 715: “function” should be “functions”
i. On page 14, line 721: ”For … f^*+\mu A_0” should be “for … f^*+\mu A_1”
j. On page 16, line 821: “this” should be “these”
k. On page 16, line 823: f(\theta_{k-1}) should be E[f(\theta_{k-1})]
l. On page 16, line 863: \log(\sqrt{n}) should be \log(n)
m. On page 17, line 867: \log(\sqrt{n}) should be \log(n)
n. On page 17, line 901: w_n\leq 1 is also used in this derivation. It should be mentioned.
o. On page 17, line 908: {… \over \rho\delta} should be {… \over \rho^2\delta}
Summary: I believe this paper presents an interesting approach in a clear and articulate manner and contains several new ideas, and could be accepted for publication if the main issues raised in the Quality section of my report is resolved.
Author Feedback

Author rebuttal: The following rebuttal significantly differs from the original one since it was mostly containing technical details concerning the first version of the paper, which can be found here http://arxiv.org/abs/1306.4650v1.

Overall, I found the reviews useful and of high quality, leading to significant improvements of the paper in terms of readability.
The changes between the first and final version of the paper are the following:
- new more ``inviting'' title.
- additional references.
- several clarifications added in the paper.
- corrections of typos that led to some confusion.
- a more ``practical'' presentation of the convergence rates, which do not require anymore an unknown learning rate.
- some precisions in the experimental section to alleviate a discrepancy between the theoretical and experimental results.
Some of the remaining reviewer's points will be addressed in a longer version of the paper (under preparation).

We also note that our C++ implementation is now included in the SPAMS software. http://spams-devel.gforge.inria.fr/